# Hearing and Hearing Loss Progression in Patients with *GJB2* Gene Mutations: A Long-Term Follow-Up

**DOI:** 10.3390/ijms242316763

**Published:** 2023-11-25

**Authors:** Aki Sakata, Akinori Kashio, Misaki Koyama, Shinji Urata, Hajime Koyama, Tatsuya Yamasoba

**Affiliations:** 1Department of Otolaryngology and Head and Neck Surgery, Graduate School of Medicine, The University of Tokyo, Tokyo 113-8654, Japan; akisakatakonishi@gmail.com (A.S.); kashioa@gmail.com (A.K.); msekiguchitymc9@gmail.com (M.K.); surata@m.u-tokyo.ac.jp (S.U.); harufsf@gmail.com (H.K.); 2Tokyo Teishin Hospital, Tokyo 102-0071, Japan

**Keywords:** *GJB2*, hearing loss, p.Leu79Cysfs*3, progressive, predictive

## Abstract

We aimed to investigate whether the degree of hearing loss with *GJB2* mutations could be predicted by distinguishing between truncating and non-truncating mutations and whether the genotype could predict the hearing loss level. Additionally, we examined the progression of hearing loss in individuals monitored for over 2 years for an average of 6.9 years. The proportion of truncating mutations was higher in patients with profound and severe hearing loss, but it was not accurate enough to predict the degree of hearing loss. Via genotype analysis, mutations of the p.Arg143Trp variants were associated with profound hearing loss, while mutations of the p.Leu79Cysfs*3 allele exhibited a wide range of hearing loss, suggesting that specific genotypes can predict the hearing loss level. Notably, there were only three cases of progression in four ears, all of which involved the p.Leu79Cysfs*3 mutation. Over the long-term follow-up, 4000 Hz was significant, and there was a trend of progression at 250 Hz, suggesting that close monitoring at these frequencies during follow-up may be crucial to confirm progression. The progression of hearing loss was observed in moderate or severe hearing loss cases at the time of the initial diagnosis, emphasizing that children with this level of hearing loss need regular follow-ups.

## 1. Introduction

Congenital hearing loss is estimated to occur in 1 in 500–1000 births, with genetic factors contributing to over 50% of cases [1]. Recent molecular genetic approaches have identified approximately 100 causative genes. Understanding each causative gene and its variants in terms of the time of onset, severity, hearing type, progression, vestibular symptoms, associated symptoms, and cochlear implant outcomes is crucial for making decisions regarding medical interventions.

Approximately 70% of all hereditary hearing loss cases are of an autosomal recessive type [1]. Autosomal recessive hearing loss is caused by mutations in both alleles, resulting in children being born with hearing loss even if their parents do not have the condition. Early and accurate hearing assessments are essential in such cases, along with guidance on coping with hearing impairment. If left untreated, congenital hearing loss can lead to poor speech development and communication difficulties. Therefore, it is necessary to compensate for hearing loss at an early stage using hearing aids or cochlear implants [2].

The age at which the indication for cochlear implantation is recommended has decreased in recent years, and earlier interventions have been shown to be more effective [3,4]. Because the recommendation for hearing aids or cochlear implants depends on the severity of hearing loss, it is crucial to determine the severity at an early stage in clinical practice. While objective assessments such as auditory brainstem response (ABR) and auditory steady-state response (ASSR) are valuable, they alone are not considered sufficient to diagnose hearing levels. Although there are reports that ABR can accurately measure hearing determination in infants [5,6,7], there are also reports emphasizing the need for behavioral assessments. Relying solely on ABR is controversial [8,9,10]. Other assessments, such as conditional orientation audiometry and visual reinforcement audiometry, which examine auditory behavior, are commonly used but are challenging to perform accurately in infants under one year of age. Consequently, genetic information has recently attracted attention as a new method to estimate hearing levels at an early age [11].

A change in genetic information occurs when a mutation occurs in the translation region, which is replaced with an amino acid sequence read by the RNA. This change is highly dependent on the site within the translational region, leading to two categories of mutations: truncating mutations for nonsense mutations, such as deletions, insertions, duplications, and changes in the termination codon, and non-truncating mutations for missense mutations, such as amino acid substitutions.

Non-truncating mutations may produce a protein product with partial functional activity. For example, a missense mutation alters different amino acids and changes the structure of the protein, though this impact is insignificant. Since a codon encodes one amino acid in units of three bases, inserting or deleting one or more amino acids in multiples of three may result in a minimal amino acid change in protein structure but not in a significant change in function.

Conversely, regarding truncating mutations, a nonsense mutation results in the termination of protein synthesis. In cases of non-3 multiple base insertions or deletions, the reading frame of the codon is shifted, and a frameshift occurs, leading to significant changes in the amino acid sequence. Most turn into stop codons after several amino acid arrangements and protein synthesis stops. In this case, major changes occur in the protein structure, resulting in a complete loss of activity or the absence of the protein itself. The degree of hearing loss depends on the degree of protein degeneration caused by the mutation and varies from mild to severe [12]. In general, truncating mutations are associated with a more severe degree of hearing impairment than non-truncating mutations [13,14].

Hearing loss caused by mutations in *GJB2* accounts for approximately half of all autosomal recessive forms of hearing loss and is the most common form of hereditary hearing loss. The carrier frequency is notably high, ranging from 2.53% to 3.73% in East Asia [15,16]. Considering a carrier frequency of 3%, there are approximately 3.6 cases of hearing loss attributed to *GJB2* mutations per 10,000 people. The severity of hearing loss due to *GJB2* mutations varies from mild to profound [17].

Gap junctions are cell–cell communication pathways formed by a ring of membrane protein hexamers called connexins, which surround channels in the membrane and are found in all tissue cells, except skeletal muscle. *GJB2* encodes connexin 26 [18], which is abundantly distributed in the cochlear ligament, fiber cells at the margins of the lacunar ligament and plate, and corticospinal supporting cells [19]. In the cochlea, connexin 26 forms a gap junction channel with connexin 30 and other proteins [20] and serves as a conduit for potassium ions between cells, maintaining intracochlear potential [21] and homeostasis so that hair cells can be excited. Mutations in the *GJB2* gene result in impaired ion cycling, leading to loss due to difficulty in maintaining inner ear homeostasis.

Over 150 *GJB2* mutations have been reported globally (https://research.cchmc.org/LOVD2/home.php (accessed on 1 August 2023)). There are racial differences in the mutation spectrum: the p.Gly12fs mutation is common in Europeans and Americans [22], p.Leu56fs is common in Ashkenazi Jewish people [23], and p.Leu79Cysfs*3 is common in East Asians [24]. In the Japanese population, the p.Gly12fs mutation, which is common in Westerners, is not found; instead, the p.Leu79Cysfs*3 mutation is found in many cases with a carrier rate of 0.658 [24,25]. People with hearing loss due to *GJB2* mutations also have variations from mild to profound hearing loss. It has been speculated that the genotype of the various genetic mutations, which vary in frequency between races, determines the degree of hearing loss [13]; however, as of yet, there are not sufficient data [26], and collecting genotype and hearing loss data from different parts of the world is needed.

Traditionally, *GJB2* mutations are associated with bilateral sensorineural non-syndromic hearing loss from birth, presumed to be non-progressive [27]. Recent interest in the progression of hearing loss has emerged [17], revealing cases of moderate hearing loss with *GJB2* mutations. Scattered reports suggest late onset [28,29], and a recent systematic review reported that 18.40% of cases show progression [17]. However, there are no reports of hearing tracking with play audiometry or standard pure-tone audiometry in large groups over long periods. Thus, detailed information in these reports, such as the initial hearing level, mutation type, and especially the course of the disease over a long time, is not known.

We, therefore, aimed to investigate whether the degree of hearing loss could be predicted by *GJB2* by distinguishing between truncating and non-truncating mutations. Additionally, we sought to establish whether the relationship between genotype and hearing loss could be predicted based on comprehensive data. Furthermore, we aimed to report the long-term hearing course with *GJB2* mutations with residual hearing and investigated the detailed background of the population with progressive hearing loss among patients with *GJB2* mutations.

## 2. Results

### 2.1. Patients

The mean age of the 75 patients at their first visit was 2.8 years old. The clinical follow-up ranged from 2 to 24 years, with an average of 6.9 years. Forty-four patients (fifty-two ears) underwent cochlear implant surgery during the follow-up period. After cochlear implantation, the follow-up hearing assessment of the implanted ear ended.

### 2.2. Genetic Diagnosis

A total of 150 allele types and the hearing level of the better ear for each allele type obtained from 75 patients diagnosed with hearing loss due to *GJB2* mutations are shown in Table 1. The most frequent allele was p.Leu79Cysfs*3 with 77 alleles (51.3%), followed by p.Gly45Glu/p.Tyr136Ter with 25 alleles (16.7%), p.Arg143Trp with 16 alleles (10.7%), c.176_191del16bp with 12 alleles (8%), and p.Val37Ile with 7 alleles (5%).

Table 2 illustrates the distribution of hearing levels in the better-hearing ear at the first evaluation among cases classified as having truncating or non-truncating allele mutations. In total, profound hearing loss was common (102/150 = 68%). The proportion of truncating mutations was higher in patients with profound and severe hearing loss (82/102 = 80% profound; 12/12 = 100% severe), whereas the proportion of non-truncating mutations was higher in patients with mild hearing loss (5/10 = 50%). There was a statistically significant difference (*p* < 0.05) in the proportion of truncating and non-truncating mutations between the profound and the mild hearing loss groups. Genotype combinations also showed a higher proportion of biallelic truncating (T/T) in profound and severe hearing loss (31/49 = 63% profound; 100% severe) and a higher proportion of biallelic non-truncating (NT/NT) in mild hearing loss (1/3 = 33%) and moderate hearing loss (2/12 = 17%) than in profound hearing loss (2/49 = 4%) (Table 3).

The degree of hearing loss in the better-hearing ear at the first visit was predominantly profound: 102 alleles (68.0%) in 51 patients with profound hearing loss, 14 alleles (9.3%) in 7 cases with severe hearing loss, 26 alleles (17.3%) in 13 patients with moderate hearing loss, and 8 alleles (5.3%) in 4 patients with mild hearing loss. When looking at the levels of hearing with a focus on each type of allele, the subjects with the p.Leu79Cysfs*3 allele had a wide range of hearing loss from profound to moderate; the subjects with the p.Arg143Trp allele had profound hearing loss, while those with the p.Val37Ile allele had moderate or mild hearing loss.

### 2.3. Longitudinal Analysis of Hearing Outcomes

The final mean age of the 46 patients enrolled in the long-term hearing follow-up was 10.5 years old, and the mean follow-up was 6.5 years, ranging from 2 to 18 years. The average age at the initial audiometry was 5.0 years old. Among the cases, three had mild hearing loss, nine had moderate, seven had severe, and twenty-nine had profound. Hearing loss progression ranged from a maximum of 7.1 dB/year for those with worsening hearing loss to −8.2 dB/year for those with conversely improving hearing loss, resulting in a wide range of results. The overall average showed a minor worsening of −0.3 dB/year. The initial mean threshold was 86.4 dB, and the last mean threshold was 86.1dB, with no significant difference.

Individually, three cases exhibited thresholds that progressed by more than 10 dB from the initial hearing evaluation to the final follow-up in four ears (3/46, 6.5%). The changes in hearing thresholds for these four ears are shown in Figure 1. One case of bilateral progression showed an increase in the hearing threshold at 5 years of age. This patient had a mutation allele of p.Leu79Cysfs*3/p.Gly45Glu/p.Tyr136Ter and already had severe bilateral hearing loss. The other two patients demonstrated disease progression in only one ear, with the p.Leu79Cysfs*3/p.Leu79Cysfs*3 mutation showcasing the fluctuating progression of hearing loss at 4–6 years of age. One patient had moderate hearing loss at the time of the initial diagnosis; however, at the time of the last examination, the hearing loss became severe. In another study, p.Leu79Cysfs*3/p.Gly45Glu/p.Tyr136Ter mutations exhibited a gradual threshold increase after the age of 4 years. When hearing loss worsened, oral steroids were administered twice at the ages of 6 and 7 years; however, these were ineffective.

### 2.4. Evaluation of Hearing Threshold Changes at Each Frequency

Figure 2a presents the threshold for each frequency and mean hearing level according to the number of years since the start of follow-up for 37 ears in 22 cases, where thresholds could be traced for all frequencies from 250 Hz to 4000 Hz at the initial evaluation. A significant difference was observed only at 4000 Hz during the follow-up period (*p* < 0.01). Figure 2b shows the threshold for each frequency and mean hearing level according to the age of the cases. No significant progression of hearing loss was observed for any frequency or mean hearing levels. The progression of hearing loss per year at mean hearing levels ranged from 7.1 dB to −8.2 dB, with an average of −0.3 dB/year, and no clear progression was observed.

Table 4 summarizes the threshold changes by frequency for all the cases. Although many cases remained unchanged at all frequencies, a number of cases worsened, and others improved. Notably, at 250 Hz, the number of worsening cases exceeded the number of improving cases, with 25% of cases experiencing worsening at this frequency (Figure 3).

## 3. Discussion

We investigated the degree of hearing loss and the long-term progression of hearing loss in patients with *GJB2* mutations based on genotype. Truncating mutations showed a strong tendency to result in profound or severe hearing loss, whereas non-truncating mutations were more likely to result in mild hearing loss. Notably, the truncating mutation p.Leu79Cysfs*3 displayed a wide range in the degree of hearing loss, while the presence of the non-truncating mutation p.Arg143Trp allele was associated with profound or severe hearing loss. These findings highlight that the degree of hearing loss cannot be solely explained by the classification of mutations into truncating or non-truncating mutations. Progression of hearing loss was observed in 4 ears in 3 cases out of 46, and when examined by frequency, the overall average did not change; however, at 4000 Hz, hearing loss was significantly worsened compared with during the follow-up period. Individually, 25% of the cases showed deterioration at 250 Hz.

### 3.1. Genetic Phenotype and Initial Hearing Levels

Protein conformational changes associated with genetic mutations are considered influential factors that determine the degree of hearing loss with *GJB2* mutations. Nonsense mutations, including frameshifts, deletions, insertions, duplications, and mutations of the splicing site, are classified as truncating mutations, while mutations such as amino acid substitutions and in-frame deletions are classified as non-truncating mutations. The degree of hearing loss decreases in the order of homozygotes for truncating mutations (T/T), compound heterozygotes for truncation/non-truncating mutations (T/N), and homozygotes for non-truncating mutations (N/N) [13,14]. In the present study, the percentage of T/T in patients with profound hearing loss was approximately 63%, while in patients with mild hearing loss, it was 33%, which is consistent with previous reports [13,14]. In the allele assessment, the proportion of truncating mutations was higher in profound and severe cases (profound: 126 allele/144 allele = 88%; severe: 27 allele/28 allele = 96%), and non-truncating mutation involvement was higher in moderate and mild cases (moderate: 10 allele/50 allele = 20%; mild: 2 allele/2 allele = 100%) [30]. However, our results and previous reports [30] have shown that profound hearing loss involves a lower percentage of truncating mutations and a higher percentage of non-truncating mutations than severe hearing loss. This suggests that predicting profound hearing loss solely through truncation or non-truncation is challenging.

One of the reasons for the higher percentage of non-truncating mutations in profound hearing loss is attributed to the impact of non-truncating mutations, specifically the p.Arg143Trp variants, which cause profound hearing loss. Functional studies have shown that p.Arg143Trp variants exert dominant-negative inhibition of connexin hemichannel activity [25]. The mutant product of the p.Arg143Trp mutation is believed to induce dominant-negative inhibition, indicating that it acts predominantly against the normal product, inhibiting its function [31,32]. In compound heterozygous p.Arg143Trp with p.Gly12fs mutations, which are associated with relatively greater hearing diversity among truncating mutations, 10 out of 10 cases presented with profound hearing loss [13]. In addition to nine out of nine cases of compound hetero with p.Leu79Cysfs*3, two out of two cases of compound hetero with p.Gly45Glu were reported to have profound hearing loss [26]. In the present study, two cases of p.Arg143Trp homozygosity and six cases of p.Leu79Cysfs*3/p.Arg143Trp compound heterozygosity, four cases of p.Gly45Glu/p.Tyr136Ter/p.Arg143Trp compound heterozygosity, and two cases of c.299-300delAT/p.Arg143Trp compound heterozygosity all showed profound hearing loss. The p.Arg143Trp mutation has been suggested as an important genotype associated with profound hearing loss.

Hearing loss associated with p.Val37Ile is generally characterized as mild–moderate [14,26,33]. Our study aligns with this pattern, revealing mild–moderate hearing loss when p.Val37Ile is present. However, in a large cohort study including 400 patients with the p.Val37Ile mutation, it was noted that homozygosity of p.Val37Ile may lead to severe–profound hearing loss to some extent. The mutation of p.Leu79Cysfs*3, the most prevalent *GJB2* gene mutation in Asia [16,24,25], has recently been linked to a diverse range of hearing loss degrees [34,35]. Similarly, p.Gly12fs, which is common in Caucasians, is associated with mild–profound hearing loss in cases of homozygosity [36]. Notably, 30% of siblings with the same genotype exhibit a hearing difference of 30 dB or more [37]. This variability has been attributed to the presence of modifier genes within the same genotype [34], and genome-wide association studies have suggested that multiple genes, rather than a single gene, are involved in hearing loss [36].

In our study, the percentage of patients with profound hearing loss and *GJB2* mutations was 68%. To date, this is one of the highest reported incidence rates. The percentage of patients with profound hearing loss among those with *GJB2* mutations varies across reporting centers, with rates of 60.8% in those undergoing cochlear implant surgery [26] and 28.8% in those who do not [27], showing a trend toward less profound hearing loss in some populations. The elevated percentage of patients with profound hearing loss in our study can be attributed to our cochlear implant facility, which tends to attract patients with profound hearing loss.

### 3.2. Progression of Hearing Loss

Hearing loss caused by *GJB2* mutations is considered non-progressive [27]. However, recent research has reported that 6.9% of patients with *GJB2*-mutation-related hearing loss pass newborn hearing screening tests [38], suggesting the possibility of acquired progressive hearing loss. A systematic review by Chan et al. also reported progression in 1,140 patients (18.4%) in 28 of 216 articles [17]. A truncating mutation in *GJB2* has been reported to cause progression at a rate of 1.409 dB/year [39]. In addition, a proposed formula for expected hearing loss progression for patients with *GJB2* mutations suggests that progression can be estimated using the formula: 3.78 + 0.96 × initial hearing level + 0.55 × years of follow-up [40]. While disease progression is reported in the studies listed in Table 5, most did not specify genotypes and comprehensive long-term follow-up results are limited. A more extensive analysis with a larger number of patients, extended observation periods, and genotype classification for each patient are crucial to determine whether differences in allele types influence the deterioration of residual hearing. Future studies with an increased number of cases and observation periods are eagerly anticipated. For this purpose, establishing a population-wide database with long-term follow-up might be warranted.

In this study, there was no discernible overall trend in the progression of hearing loss, indicated by an annualized rate of −0.3 dB/year. However, individual case analyses revealed that in three cases with four ears, hearing loss progressed by 10 dB or more until the last observation, suggesting the presence of progressive sensorineural hearing loss caused by *GJB2* gene mutations. A more detailed evaluation of the progression of hearing loss by frequency revealed that there was a significant threshold shift at 4000 Hz during the follow-up period. When we examined each case across all frequencies, there was a constant improvement in the thresholds in some of the cases, alongside instances of unchanged and worsening hearing loss. The observed improvement in thresholds was attributed to improvement in test accuracy with age, indicating challenges in hearing assessments for young children [42]. Several factors could contribute to the improvement in auditory sensitivity with age, such as (1) attentional factors, (2) learning effects, (3) earphone leakage, (4) differences in ear canal size, (5) middle-ear disorders in younger children, (6) age-dependent interactions with the test procedure used to measure thresholds, and (7) a true difference in auditory sensitivity as a function of age. Although regular checks of the ear drum were conducted, routine tympanograms were not performed, leaving the possibility of middle-ear disorders unaddressed in this population, a limitation of the study. However, in cases showing worsening hearing loss, meticulous examination of the tympanic membrane with a microscope and tympanometry was performed to rule out conductive hearing loss. Despite some observed improvement, the number of cases with a progression of 10 dB or more exceeded the number of cases with threshold improvement at 250 Hz. This suggests that some cases of *GJB2* mutations exhibit an increase in the threshold, especially at 250 Hz. Therefore, close monitoring at these frequencies during follow-up may be crucial for confirming progression. A report examining eight cases also demonstrated a gradual progression of hearing loss at middle–high and low frequencies, which is consistent with the present study findings and previous reports [32].

### 3.3. Hearing Level at Initial Examination in Cases of Progressive Hearing Loss

In our study, deterioration was observed in cases with the p.Leu79Cysfs*3 mutation, and no progression was observed in patients with the p.Val37Ile mutation. In contrast, Lee et al. [26] suggested that the p.Val37Ile mutation caused rapid and significant deterioration, whereas p.Leu79Cysfs*3 caused minimal deterioration. Chen et al. also suggested that hearing progression occurs in p.Val37Ile patients. The discrepancy in these results may be due to the initial hearing levels of the patients. In our cases, patients with p.Val37Ile mutations exhibited mild hearing loss, whereas other reports indicated moderate or severe hearing loss in such patients. For homozygous mutations of p.Val37Ile, hearing loss is more likely to progress in patients with severe–moderate hearing loss than in patients with mild hearing loss [32]. In the case of mitochondrial A3243G-related hearing loss, rapid progression may occur after hearing loss has progressed to 55 dB or more [43]. In our study, the three cases with threshold progression initially had moderate hearing loss in one case and severe hearing loss in two cases. This suggests that progression may be more dependent on the hearing level, reflecting inner ear damage, rather than the type of allele. The more impaired the inner, the more vulnerable it may be to environmental factors. The more severe the hearing loss, the greater the need for amplification, which may cause acoustic trauma.

One notable limitation of this study is the insufficient total number of cases for robust statistical analysis. Although the frequency-by-frequency evaluation showed that there were a few cases with threshold elevation at 250 Hz, the number was too small to be meaningful. Although we have highlighted that the p.Arg143Trp mutation is indicative of severe hearing loss, the relationships between other genotypes and hearing levels cannot be adequately described from this study alone because of the limited sample size for each genotype. Similarly, the observed trend toward progressive hearing loss due to p.Leu79Cysfs*3 cannot be fully supported by this study alone. However, we anticipate that the emerging results of analogous studies in the future will contribute to a clearer understanding of the relationship between genotype and the degree of hearing loss, as well as the risk of hearing loss progression. In this context, we believe it is worthwhile to report the results of this study. Another limitation lies in the relatively short observation period for some cases. Although the average follow-up period of 6.9 years is considered relatively long compared with other reports, a subset of cases was tracked for only approximately 2 years. We believe that the progression of hearing loss needs to be evaluated over a longer period, and it is imperative to continue monitoring these cases over an extended timeframe to thoroughly analyze their long-term prognosis. Additionally, we decided to exclude the data of ABR and ASSR in favor of relying solely on behavioral audiometry data because the evaluation of accurate hearing levels with ABR and ASSR remains controversial. This decision introduces another limitation to our study. Consequently, this approach may have led to a potential oversight of cases with progression occurring within the first year after birth. Future studies with more extensive datasets and prolonged observation periods should consider integrating multiple assessment methods for a comprehensive understanding of hearing progression.

## 4. Materials and Methods

### 4.1. Patients

Patients with hearing loss who visited the Department of Otolaryngology and Head and Neck Surgery at the University of Tokyo Hospital between 2009 and 2022 were tested for *GJB2* mutations. Seventy-five patients (thirty-seven females and thirty-eight males) with homozygous or compound heterozygous *GJB2* mutations were identified. The subjects included 7 adults, 9 pediatric cases, and 59 infants. The genotype, degree of hearing loss, and progression of hearing loss were retrospectively examined. All procedures were performed in accordance with the Declaration of Helsinki and approved by the University of Tokyo Human Ethics Committee (no. 3543-3). All the patients provided informed consent for the use of their clinical data. If patients were less than 20 years old, informed consent was obtained from their parents.

### 4.2. Audiological Evaluations

The level of the better-hearing ear at each frequency was assessed with behavioral audiometry at the time of the first visit. In cases where the first visit was before the age of 1 year, hearing was assessed after the age of 1 year when behavioral audiometry shows relatively consistent results. Hearing was assessed separately for the left and right ears using standard pure-tone audiometry or play audiometry (for subjects using headphones). The degree of hearing loss was calculated as the mean hearing levels ([500, 1000, 1000, + 2000 Hz]/4). Profound hearing loss was defined as 90 dB or more, 70 dB to 90 dB as severe, 40 dB to 70 dB as moderate, and 25 dB to 40 dB as mild hearing loss. The proportion of truncating mutations and non-truncating mutations between profound and mild hearing loss was compared using Fisher’s exact test. Tympanic findings were evaluated by an otolaryngologist, but tympanometry was not routinely performed except in cases where hearing loss had progressed. In cases of progressive hearing loss, a close examination of the tympanic membrane with a microscope and tympanometry was necessary to rule out conductive hearing loss.

### 4.3. Longitudinal Analysis of Hearing Outcomes

The progression of hearing loss was examined in 46 patients (92 ears) monitored for more than 2 years since infancy, using play audiometry or standard pure-tone audiometry. If the mean hearing loss increased by 10 dB or more from the first examination to the last examination, the case was recorded as progressed hearing loss. In addition, the differences between initial and last hearing evaluations were compared using paired t-tests, and the hearing loss progression rate was calculated as the change in hearing loss from the last observation to the first examination in each ear divided by the number of years observed.

For 36 ears of 22 patients with complete hearing data from 250 to 4000 Hz at the time of the initial examination, the presence or absence of hearing changes at each frequency were individually evaluated. Hearing and mean hearing levels for each frequency in each ear were categorized according to the follow-up period and age, and their trends were examined. In addition, differences between the initial hearing evaluations and last evaluations were compared using a one-way analysis of variance, and cases with an increase in the threshold by 10 dB or more from the initial hearing threshold to the last examination were categorized as worsening cases, and cases with a decrease in the threshold of 10 dB or more were categorized as improving cases, as reported previously [41]. The remaining cases were unchanged.

### 4.4. Genetic Testing

The Sanger sequencing method was performed for patients who underwent genetic testing between 2009 and 2011. The invader method [44] has been used since 2012, and in 2015, the gene was detected using next-generation sequencing. Mutant alleles found in homozygous or compound heterozygous *GJB2* mutations were classified as truncating or non-truncating mutations, respectively. The genotypes representing the combinations of mutant alleles were classified and evaluated for each.

## 5. Conclusions

This study investigated the genotypes, degree of hearing loss, and the long-term progression of hearing loss in cases of *GJB2* mutations. Most patients with hearing loss due to *GJB2* mutations displayed profound hearing loss at the initial diagnosis (68%). The accurate prediction of hearing levels proved challenging using classifications based on truncating and non-truncating mutations. Genotype analysis revealed that mutations of the p.Arg143Trp variants were associated with profound hearing loss, while mutations of the p.Leu79Cysfs*3 allele exhibited a wide range of hearing loss, suggesting that specific genotypes may serve as predictors for hearing levels. Progressive cases were identified in 6.5% of patients via long-term observation, with the mutation of p.Leu79Cysfs*3 appearing to be linked to progression. On a frequency-by-frequency assessment, there was a significant difference at 4000 Hz and some tendency for progression at 250 Hz, suggesting that close monitoring at this frequency during follow-up may be important to confirm progression. Progression of hearing loss was observed in moderate or severe hearing loss cases at initial diagnosis, emphasizing the need for regular follow-ups in children with these levels of hearing loss.

## Figures and Tables

**Figure 1 ijms-24-16763-f001:**
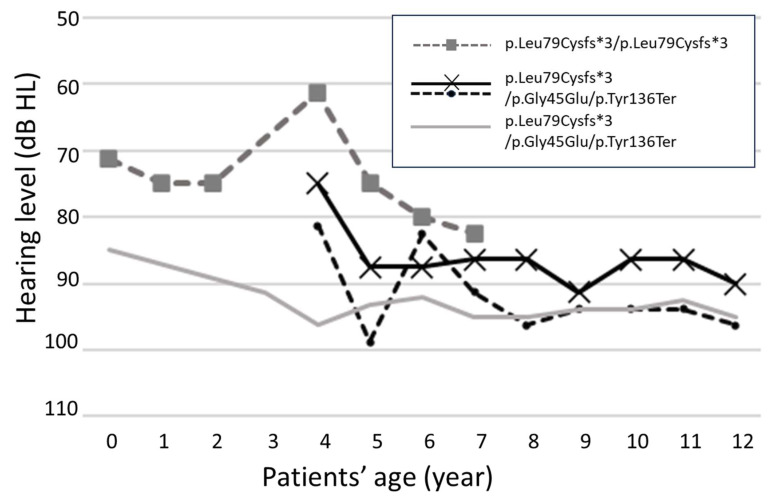
Hearing changes in 4 ears in 3 cases with progressive hearing loss. Threshold progression of more than 10 dB from initial hearing evaluation to final follow-up in 4 ears. Black and black dotted lines indicate the same patients.

**Figure 2 ijms-24-16763-f002:**
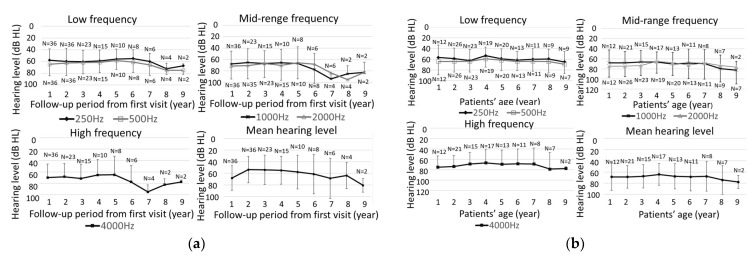
(**a**) Thresholds for each frequency and quadrant average according to the number of years since the start of the follow-up observation. The relationship between the hearing threshold (*y*-axis) and the number of years of follow-up (*x*-axis) was averaged only for cases that could be followed for more than one year. Hearing changes were evaluated at 250 Hz, 500 Hz, 1000 Hz, 2000 Hz, 4000 Hz, and quadrants. N is case number. Date indicates mean ± SEM. (**b**) Frequency and quartile mean hearing thresholds for each case age. Date indicates mean ± SEM.

**Figure 3 ijms-24-16763-f003:**
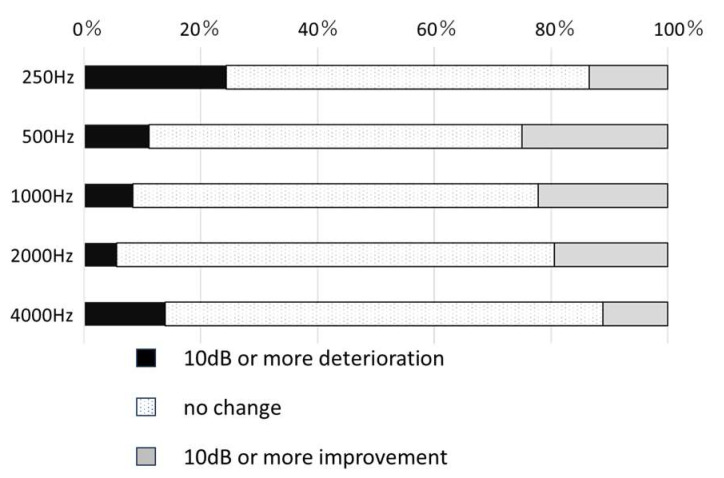
Threshold changes by frequency. Expressed as the number of ears that worsened by more than 10 dB, remained unchanged, or improved by more than 10 dB threshold at the point of the initial frequency-specific hearing test and at the last examination.

**Table 1 ijms-24-16763-t001:** Genotypes and degrees of hearing loss at initial diagnosis.

Variation	NT/T	Number of Alleles	Hearing Level of Better Ear at First Visit			
			Profound	Severe	Moderate	Mild
			≧90 dB	70–90 dB	40–70 dB	25–40 dB
p.Leu79Cysfs*3	T	77	48	10	19	
p.Gly45Glu p.Tyr136Ter	T	25	18	2	3	2
c.176_191del16bp	T	12	9	1	1	1
c.299-300delAT	T	6	5	1		
p.Typ134Ter	T	1	1			
p.Ala171Ter	T	1	1			
p.Arg143Trp	NT	16	16			
p.Val37Ile	NT	7			2	5
p.Thr86Arg	NT	3	2		1	
p.Arg127Cys	NT	2	2			
Total		150	102 (102/150 = 68.0%)	14 (14/150 = 9.3%)	26 (26/150 = 17.3%)	8 (8/150 = 5.3%)

T: truncating mutation; NT: non-truncating mutation.

**Table 2 ijms-24-16763-t002:** Degrees of hearing in 75 people with *GJB2* gene mutations. Alleles were classified as truncating or non-truncating.

	Number of Alleles	Hearing Level of Better Ear at First Visit			
		Profound	Severe	Moderate	Mild
Truncating mutation	122	82	12	23	5
		82/102 = 80%	12/12 = 100%	23/26 = 88%	5/10 = 50%
Non-truncating mutation	28	20	0	3	5
		28/102 = 27%		3/26 = 12%	5/10 = 50%
Total	150	102	12	26	10

**Table 3 ijms-24-16763-t003:** Degrees of hearing in 75 people classified with genotypes.

	Number of Cases	Hearing Level of Better Ear at First Visit			
		Profound	Severe	Moderate	Mild
T/T	52	31	11	9	1
		31/49 = 63%	11/11 = 100%	9/12 = 75%	1/3 = 33%
T/NT	18	16	0	1	1
		16/49 = 33%		1/12 = 8%	1/3 = 33%
NT/NT	5	2	0	2	1
		2/49 = 4%		2/12 = 17%	1/3 = 33%
Total	75	49	11	12	3

T/T: truncating/truncating; T/NT: truncating/non-truncating; NT/NT: non-truncating/non-truncating.

**Table 4 ijms-24-16763-t004:** Threshold changes by frequency.

Frequency (Hz)	250	500	1000	2000	4000
10 dB or more deterioration	9	4	3	2	5
No change	23	24	26	28	28
10 dB or more improvement	5	9	8	7	4
Total (ear)	37	37	37	37	37

**Table 5 ijms-24-16763-t005:** Previous reports of progressive hearing loss with *GJB2* mutations.

Author	Publication Year	Criteria of Progression	Number of Cases	Progression Cases	Ratio	Follow-Up Period
Chan et al. [33]	2010	5 dB deterioration or 1 dB/year	33	8	8/33 = 24%	5.3-year average
Tsukada et al. [26]	2010		26	4	4/26 = 15%	2 years or more
Kenna et al. [14]	2010	10 dB or more deterioration at 2 frequencies 15 dB or more deterioration at a frequency of	84	47	47/84 = 56%	13-month average
Imai et al. [30]	2013	1 dB/year or more	32	1	1/32 = 3%	1–24 years
Chan et al. [17]	2014	(Systematic review)			18.70%	
Nakano et al. [41]	2020	10 dB or more deterioration	34	4	4/34 = 12%	1–17 years
Chen et al. [40]	2020	0.5 dB/year or more	227	38	38/227 = 17%	0.3~21 years
Sakata	This paper	10 dB or more deterioration	48	3	3/48 = 6%	2–24 years

## Data Availability

The data presented in this study are available on request from the corresponding author.

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
