# Peer review of "Hearing and Hearing Loss Progression in Patients with GJB2 Gene Mutations: A Long-Term Follow-Up"

_ijms, 2023, doi:10.3390/ijms242316763_

Round 1

Reviewer 1 Report

Comments and Suggestions for Authors

I believe this is a relatively interesting manuscript on the hearing loss and hearing loss progression in relation to the various mutations of the connexin gene. 

English language needs improvement. apart form syntax and grammatical errors, more concise use of the language is needed especially in the abstract and introduction sections and more particular in the paragraph that describes truncating and non-truncating mutations.   

Some references are missing for example in the introduction there where the authors state that genetic information has attracted attention as a method to estimate hearing levels

In practical terms a few years period may not be sufficient for the follow up in relation to hearing loss. 

They encountered some improvements. Maybe they have included some conductive hearing losses. They do not specify if they have checked only sensorinural hearing losses. 

they do not report whether their results have reached statistical importance. Maybe a chi-square test is sufficient in their case. 

their conclusions are important but their manuscript in general needs improvement 

Comments on the Quality of English Language

English language needs improvement. apart form syntax and grammatical errors, more concise use of the language is needed especially in the abstract and introduction sections and more particular in the paragraph that describes truncating and non-truncating mutations. 

Reviewer 2 Report

Comments and Suggestions for Authors

General comments:

The study aimed to explore the connection between the GJB2 genetic variant and the severity or progression of hearing loss. The authors examined 75 patients initially and monitored 46 patients over two years. They found a higher occurrence of severe hearing loss in cases with specific genetic mutations. Interestingly, despite a small proportion, some patients showed progression in hearing loss, particularly at 250 Hz frequency. The study seems to suggest that confirming the GJB2 genotype can be valuable in understanding hearing loss characteristics. However this is not clearly stated. The strong point of the study is the long observation period of a group of subjects. On the other hand the weakness is that study lacks clearly defined purpose and clearly provided conclusions.

Specific comments:

Introduction

L41-43 Please provide reference e.g.:

Leigh, J., Dettman, S., Dowell, R., & Briggs, R. (2013). Communication development in children who receive a cochlear implant by 12 months of age. Otology & Neurotology, 34(3), 443-450.

L47-49 „Objective assessments such as auditory brainstem response and auditory steady state response alone are not considered sufficient to diagnose hearing levels.” - The sentence is very controversial in my opinion. Please provide some references. Auditory brainstem responses are indeed to estimate hearing thresholds in children with great success, e.g.:

Sininger, Y. S. (1993). Auditory brain stem response for objective measures of hearing. Ear and Hearing, 14(1), 23-30.

Hang, A. X., Roush, P. A., Teagle, H. F., Zdanski, C., Pillsbury, H. C., Adunka, O. F., & Buchman, C. A. (2015). Is “no response” on diagnostic auditory brainstem response testing an indication for cochlear implantation in children?. Ear and hearing, 36(1), 8-13.

Please provide more clear rationale for the study. What we do know, what is unknown? How this study wants to surpass the previous studies? What number of the subjects is needed to see the effect?

Material and Methods

L126-128 “however, in cases where the first visit was before the age of 1 year, hearing was assessed after the age of 1 year, as an accurate assessment of hearing was not possible due to monthly developmental factors in the response to sound.” – I do not understand this sentence. It is certainly possible to do some testing in children below 1 year, e.g. otoacoustic emissions or auditory brainstem responses.

Where there any tests of middle ear status, e.g. tympanometry?

Results

L202-204 This sentence could be better formulated.

L230-232 Was the worsening of hearing a statistically significant effect?

Discussion

L330-331 “there was a constant improvement in the thresholds in some of cases, in addition to unchanged and worsening hearing loss.” Please explain it better, there cannot be improvement in thresholds simultaneously with worsening hearing loss.

Please consider adding information about limitations of the study and some ideas for future research.

Conclusions

These are not conclusions. This is rather a summary of obtained results. Please add some concluding remarks.

Editing errors:

Figure 1. The y axis caption should be rotated by 180 degrees.

L374 I guess the authors wanted to say: higher IN patients with profound loss

Round 2

Reviewer 1 Report

Comments and Suggestions for Authors

I believe that the authors  responded to all my comments adequately  

Comments on the Quality of English Language

authors report that they have utilized an "english review company"

Reviewer 2 Report

Comments and Suggestions for Authors

I thank the authors for introducing suggested comments.